# Spatiotemporal Characteristics of the Coupled Coordination Degree of Ecosystem Services Supply and Demand in Chinese National Nature Reserves

**DOI:** 10.3390/ijerph20064845

**Published:** 2023-03-09

**Authors:** Cheng Huang, Jie Zeng, Wanxu Chen, Xinyu Cui

**Affiliations:** 1Department of Geography, School of Geography and Information Engineering, China University of Geosciences, Wuhan 430074, China; 2Hubei Key Laboratory of Regional Ecology and Environmental Change, Wuhan 430074, China; 3Key Labs of Law Evaluation of Ministry of Natural Resources of China, Wuhan 430074, China; 4State Key Laboratory of Earth Surface Processes and Resource Ecology, Beijing Normal University, Beijing 100875, China

**Keywords:** ecosystem services supply, ecosystem services demand, kernel density estimation, coupling coordination model, Chinese national nature reserves

## Abstract

Nature reserves (NRs) are the main components of protected areas and geographic spaces, with unique natural and cultural resources. The establishment of nature reserves has not only strengthened the protection of specific species but has also played a vital role in the protection of ecosystem services (ESs). However, few studies have been conducted to systematically assess the effectiveness of nature reserves from the perspective of ecosystem services supply and demand (S&D) or make comparisons between the conservation effects of different types of nature reserves. This study analyzed the spatiotemporal characteristics of ecosystem service supply and demand in 412 Chinese national nature reserves. The results showed that both supply and demand for ecosystem services per unit area show a spatial pattern of increasing from west to east. The supply–demand matching pattern is dominated by high supply–high demand (H–H) and low supply–high demand (L–H) in the central and eastern regions, and high supply–low demand (H–L) and low supply–low demand (L–L) in the northeast, northwest, and southwest regions. The coupling coordination degree (CCD) of ecosystem services supply and demand increased from 0.53 in 2000 to 0.57 in 2020, and the number of NRs reaching the coordinated level (>0.5) increased by 15 from 2000 to 2020, representing 3.64% of the total number of protected areas. Steppe meadows, ocean coasts, forest ecosystems, wildlife, and wild plant types of nature reserves all improved more obviously. This provides a scientific basis for strengthening the ecological and environmental supervision of nature reserves, and the research methods and ideas can provide references for similar research.

## 1. Introduction

China plays an important role in global biodiversity and ecosystem services (ESs) conservation by establishing different categories of protected areas, such as geological sites, scenic spots, national parks, forest parks, and nature reserves (NRs) [1]. As of 2018, the number of different levels of NRs in China reached 2750, with a total area of 1.47 million km^2^. NRs are the main part of protected areas, which are geographical spaces with unique natural and cultural resources in which specific protection measures and related regulations are implemented to reduce human disturbance and achieve the long-term protection of the ecological environment and ESs [2]. In 2010, the Aichi Biodiversity Targets of the Convention on Biological Diversity highlighted the protection of ESs as one of the main objectives of establishing NRs [3]. The protection of ESs has gradually become an important consideration in NRs policy development [4]. It provides an important basis for optimal management strategies for NRs. However, due to rapid socio-economic transformation, China’s ecological environment is under enormous pressure [5,6,7]. The long-term sustainability of NRs faces unprecedented challenges [8]. Systematic assessment of ESs conservation effectiveness of China’s NRs is important for the formulation of conservation policies and regional ecological management.

The ecosystem services of NRs are of concern to a wide range of scholars [9,10], and have been explored in several studies [11,12]. Hugé et al., based on a combination of a literature review and a user feedback survey, developed ecosystem service assessment tools applicable to the African biosphere reserve context to classify and effectively assess ecosystem services [13]. Mahlalela et al. used the Q methodology to identify and analyze the diverse perspectives held by different stakeholders about the Hawane Dam and Nature Reserve wetland ecosystem services (ESS) and believed that stakeholders hold three different views [14]. SrikantaSannigrahi et al. used supervised machine learning methods to measure the spatial and temporal variability of 17 key ESs in the Sundarbans Biosphere Reserve, India, arguing that mangroves and water bodies are highly sensitive to any human or natural encroachment [15]. In terms of the conservation effectiveness of nature reserves, previous studies have assessed the conservation effectiveness of forest ecological NRs in terms of changes in forest cover [16,17]. Some scholars confirmed the conservation effects of inland wetland NRs from changes in wetland area [18] and net primary productivity [19]. Desert ecosystems and steppe meadow NRs were discussed in terms of combating desertification and preventing grassland degradation, respectively [20,21]. It was concluded that desertification and grassland degradation in NRs have slowed down [22]. Although various studies have reviewed the conservation effectiveness of NRs based on individual cases or single types from the perspective of ESs [1,20], few studies have been conducted to systematically assess NRs’ effectiveness from the perspective of ESs supply and demand (S&D) and lack comparisons of the conservation effects of different types of NRs.

ESs refer to the benefits that humans obtain directly or indirectly from an ecosystem [23,24]. Research on the relationship between ESs and human well-being is becoming a hot topic in the study of ESs [25]. The Millennium Ecosystem Assessment Report noted that >60% of global ESs showed a downward trend. Imbalances caused by reduced supply and increased demand for ESs are considered a potential cause of ecosystem degradation [26]. Most research has focused on the supply of ecosystem services, while overlooking the needs of human society. However, the concept of ecosystem services was originally centered around humans and aimed at promoting human well-being and achieving sustainable development. Therefore, only by effectively combining the supply capacity of natural ecosystem services with the needs of human society can the social value of research outcomes be enhanced [27]. Under the influence of human activities, the value generated by ecological products does not fully match the products of ecological services required by humans, which easily leads to spatiotemporal dislocation [28,29]. As a special carrier of ESs, NRs have relatively complete ecological structures, processes, and functions. However, the coupled coordination mechanism of ESs S&D is not clear. The coupled coordination degree (CCD) model can assess the degree of coordination between different subsystems and provides an effective method for the study of ESs S&D [30]. Therefore, it is necessary to further analyze the characteristics of ESs changes in NRs and identify ESs issues from the perspective of matching the S&D and CCD of ESs.

Measuring the CCD of ESs S&D in NRs and matching situations helps to effectively manage ecosystems, rationally allocate natural resources, and maintain ecological security [31]. In this study, 412 national NRs in China were used to analyze the spatial match between S&D of ESs and the coupling coordination characteristics based on land use/land cover (LULC) data and socio-economic data in 2000, 2010, and 2020. This study aimed to shed light on the following two issues: (1) to quantify the spatiotemporal distribution characteristics and pattern of matching of ESs S&D in NRs; (2) to assess the conservation effectiveness of different types of NRs on the CCD of ESs S&D. This provides an important scientific basis for strengthening ecological and environmental supervision of NRs and achieving sustainable development.

## 2. Data Source and Methods

### 2.1. Study Area

The establishment of NRs is considered an important means of preserving biodiversity and mitigating human–land conflicts [32]. It plays an irreplaceable role in protecting the ecosystem. In this study, 412 Chinese national NRs were used as the study area, and the land area of the reserves was about 976,200 km^2^. The spatial distribution of the NRs is shown in Figure 1. According to different protection objects and purposes, the NR system is divided into three categories: natural ecosystem NRs (270), wildlife NRs (120) and natural relics NRs (21), which can be subdivided into nine subtypes: forest ecosystems (187), steppe meadows (4), desert ecosystems (13), inland wetlands (48), ocean coasts (18), wildlife (106), wild plants (14), geological relics (14) and ancient biological relics (7). The regional distribution of NRs varies markedly, with a relatively small number of reserves in the west and a relatively large area of individual reserves, up to 300,000 km^2^. A relatively dense concentration of reserves exists in the center and south, with a relatively small area of individual reserves, as small as 1.2 km^2^.

### 2.2. Date Sources

The LULC data, normalized difference vegetation index (NDVI), population density spatial distribution dataset, and GDP density spatial distribution dataset used in this study were all obtained from the Resource and Environment Science and Data Center (http://www.resdc.cn, accessed on 1 January 2022). Among them, the LULC dataset uses Landsat–MSS, Landsat–TM/ETM and Landsat8 remote sensing image data as the information source, and after image alignment, accuracy correction, and stretching processing, the 1 km land use raster was obtained through a manual decoding method. LULC types include 6 primary types (farmland, woodland, grassland, water bodies, construction land, and unused land) and 25 secondary types. Since Xie et al. [33] emphasized the importance of wetlands in ESs, this study referred to previous research results [30,34] to reclassify the LULC dataset into seven categories: farmland, woodland, grassland, water bodies, wetland, construction land, and desert for the study.

### 2.3. Methods

#### 2.3.1. Accounting for Supply and Demand of Ecosystem Services

After Costanza et al. [23] proposed a classification of ESs and an equivalence factor, Xie et al. [33] constructed a Chinese ESs value equivalent per unit area by adjusting the ESs equivalence coefficient based on the actual situation in China. The equation for calculating the ESs supply is as follows:(1)$$ESS=\sum_{k=1}^{m}\sum_{i=1}^{n}\left(VC_{i,k}\times A_{i}\right)$$
where *ESS* denotes ESs supply; *VC_i,k_* is the equivalent of the *k*-th ESs of the *i*-th LULC; *m* and *n* denote the number of categories of ESs and the number of LULC, respectively.

The equivalent factor method is effective in estimating ESs supply at different scales and regions, but the ESs value provided by the same LULC also varies [35,36]. In this study, the *NDVI* dataset was used to revise the ESs supply estimated by the ESs value equivalent table, to finally obtain the ESs supply in Chinese national NRs in China. Since land use types such as water bodies, wetland, construction land, and desert are sparsely vegetated and their *NDVI* values are basically negative, only farmland, forest land and grassland were corrected for vegetation cover here [37,38]. The *NDVI* revision factor was calculated to range from 0 to 4.18.
(2)$$f_{ij}=\frac{NDVI_{f}-NDVI_{\min}}{NDVI_{\max}-NDVI_{\min}}$$
(3)$$ESS_{\mathrm{correct}}=ESV\times \frac{f_{\mathrm{ij}}}{\bar{f_{j}}}$$
where *NDVI_f_* is the annual average normalized vegetation index of grid *f*; *NDVI_max_* and *NDVI_min_* denote the maximum and minimum values of the annual average normalized vegetation index; *f_ij_* is the vegetation cover of ecosystem type *j* in cell *i*; $\bar{f_{j}}$ is the average vegetation cover of ecosystem type *j* in the study area; *f_i_* is the vegetation revision factor of cell *i*; *ESS_correct_* is the corrected ESs supply.

ESs demand is for specific products and services that are consumed or used within a certain spatial and temporal context [39], including actual and potential demand [40,41]. The mainstream methods for measuring ESs supply are public participation methods [42] and indicator methods [43,44]. The public participation method is susceptible to factors such as experts’ experience, perceptions, preferences, and attitudes, and the evaluation results are highly subjective [45]. Therefore, in this study, the land use intensity, population density, and GDP density were used as measures to represent the land demand, population demand, and economic demand in NRs, respectively [46]. Among them, the criteria for classifying the land use intensity refer to the study of Zhuang et al. [47] and classify the land use intensity into four classes (Table 1).

As a result of the large differences in population density and GDP density between NRs, the data were logged to mitigate the effects of sharp fluctuations. The measurement formula is as follows:(4)ESD=LDI×lnP×lnG
(5)$$LDI=100\times \left(\sum_{i=1}^{n}P_{i}\times Q_{i}\right)$$
where *ESD* is the ESs demand of the NRs; *P* represents population density (person/km^2^); *G* is economic density (CNY/km^2^); *LDI* is the land use intensity index; *P_i_* represents land use intensity class *i*; *Q_i_* is the proportion of the area of land use intensity *i* to the area of the corresponding NRs; *n* is the number of land use intensity gradations in the study area, here 4.

#### 2.3.2. Ecosystem Services Supply and Demand Matching and Coupling Coordination Degree

This study used NRs as the basic research unit to analyze and measure the matching pattern of ESs S&D and the CCD of different NRs. The ESs S&D in NRs were first standardized by z-score separately, and a two-dimensional coordinate system was constructed with the standardized demand as the *x*-axis and the standardized supply as the *y*-axis to obtain the coordinate points (*x*,*y*) of the NRs. Among them, the first to fourth quadrants represent high supply–high demand (H–H), high supply–low demand (H–L), low supply–low demand (L–L) and low supply–high demand (L–H), respectively. The model expressions are as follows:(6)$$x=\frac{x_{i}-\bar{x}}{s}$$
where *x* is the z-score standardized supply or demand; *x_i_* is the supply or demand of ESs in the *i*-th nature reserve; $\bar{x}$ represents the average of the observed values; and *s* is the study area standard deviation. 

The coupling degree (Equation (8)) indicates the strength of the association between the dimensions of the system [48]. However, it does not reflect whether the dimensions are harmoniously developed [49]. The CCD (Equation (10)) can be a good solution to this problem. To eliminate the influence brought by the magnitude, the S&D of ESs were first standardized. The CCD model was introduced to explore the coupled coordination relationship between the S&D of ESs and NRs.
(7)$$X=\frac{x_{i}-x_{\min}}{x_{\max}-x_{\min}}$$
(8)$$C=2\times \sqrt{\frac{X_{S}\times X_{D}}{{\left(X_{S}+X_{D}\right)}^{2}}}$$
(9)$$T=\alpha \times X_{S}+\beta \times X_{D}$$
(10)$$D=\sqrt{C\times T}$$
where *X* is the supply or demand of ESs after standardization of NRs; *x_i_* is the supply or demand of ESs in the *i*-th nature reserve; xmax is the maximum value of NRs; *x_min_* is the minimum value of NRs; *C* is the coupling degree; *T* is the comprehensive coordination index of S&D; *X_S_* and *X_D_* denote ESs supply and ESs demand, respectively; *α* and *β* are the corresponding weights, which are taken as 0.5 because the importance of ESs S&D is the same. Referring to Han et al. [39], the CCD of ESs S&D was classified into the following eight levels: (1) *D* ∈ [0,0.2), the ESs S&D are severely dysfunctional; (2) *D* ∈ [0.2,0.3), the ESs S&D are moderately dysfunctional; (3) *D* ∈ [0.3,0.4), the ESs S&D are mildly out of balance; (4) *D* ∈ [0.4,0.5), the ESs S&D are on the verge of being out of balance; (5) *D* ∈ [0.5,0.6), the ESs S&D are basically coordinated; (6) *D* ∈ [0.6,0.7), the ESs S&D are mildly coordinated; (7) *D* ∈ [0.7,0.8), the ESs S&D are moderately coordinated; (8) *D* ∈ [0.8,1], the ESs S&D are well coordinated.

#### 2.3.3. Kernel Density Estimation

The kernel density function, a nonparametric estimation method for studying the distribution characteristics of data samples, derives a variety of different estimation functions. Its core is to use a smoothed peak function to fit the CCD, in order to simulate the real probability distribution curve of the CCD of ESs S&D. By comparing the estimated CCD kernel density in different years, it can reflect the overall trend of CCD in NRs.
(11)$$\hat{f}(x)=\frac{1}{n\delta }\sum_{i=1}^{n}k(\frac{X_{i}-x}{\delta })$$
where the function *k(·)* is the kernel function, i.e., the weight function in the density function estimation; the larger the specific value of the bandwidth parameter *δ*, *X_i_-x* is the distance from the estimated point *x* to the event *X_i_*. The larger the neighborhood in the vicinity of *x*, the smoother the density function will be, so the bandwidth parameter *δ* is also called the smooth parameter. Commonly used kernel functions include Gauss, Epanechnikov, rectangle, etc. [50]. Since the Epanechnikov kernel function can achieve the optimum of the integration squared error for a given sample analysis and reduce the efficiency loss of data fitting, the Epanechnikov kernel function is used in this study as follows:(12)$$k(\psi )=\frac{3}{4}(1-\psi^{2})\left\{\left|\psi \right|\leq 1\right\}$$

## 3. Results

### 3.1. Spatiotemporal Characteristics of Ecosystem Services Supply and Demand

Based on LULC and NDVI data from 2000 to 2020, we quantified the supply of the four ESs by different land use types in NRs through Equations (1)–(3) (Table 2). The total value of ESs in NRs showed a conservation pattern of initially decreasing and then increasing, from USD 100,381.83 million in 2000 to USD 99,834.51 million in 2010, and then increasing to USD 101,425.38 million in 2020, a rise of 1.04% in 20 years. In terms of different land use types, woodlands and grasslands provided the highest ESs supply, with the sum of the two accounting for 61.92% of the total value of ESs in 2000 (Figure 2), while this share decreased to 52.97% by 2020. This is mainly due to the conversion of a large amount of woodlands and grasslands to wetlands and waters from 2000 to 2020, which reduced the value of ESs provided by woodlands and grasslands. Wetlands and water bodies, as land use types with a higher value per unit of ecosystem, showed a significant increase in the value of ESs provided. Wetlands and water bodies decreased from USD 17,052.57 and USD 9419.12 million in 2000, to USD 22,415.50 and USD 12,631.61 million in 2020, respectively, and their shares of total ESs increased by 5.11% and 3.07% of total ESs, respectively. In terms of ESs, the value supply of ESs showed a ranking of regulating services > supporting services > provisioning services > cultural services. Provisioning and support services have declined since 2000, regulating and cultural services have increased, with the four ESs shifting from 6.42%, 26.92%, 58.54% and 8.13% of total ESs in 2000 to 5.98%, 24.64%, 61.05% and 8.33% in 2020, respectively.

In terms of spatial distribution, there was a clear spatial heterogeneity in the S&D of ESs (Figure 3). In terms of supply, the highest value of ESs per unit area was USD 3204 per ha. The overall spatial pattern was high in the east and low in the west. The western region has an arid climate, and the reserve is dominated by land cover types with low ESs values per unit area, such as unused land and grassland. Due to the high altitude of the Qinghai–Tibet Plateau, there are some glacial lakes and wetlands, so there are a few scattered areas with high ESs values. The central and southern NRs had a higher overall ESs unit value compared to the western region, due to the wet climate and the distribution of woodland wetlands. In the northeast, due to long winters and a wet climate, there are many swampy wetlands in the NRs, and the ESs values are relatively high. 

Using Equations (4) and (5), we measured the and in the NRs. The overall demand for ESs was also high in the east and low in the west, with a spatial pattern increasing from inland to coastal (Figure 3). The high latitude, poor climatic conditions, and low human and economic activity in the northeast also resulted in lower demand for ESs. Western regions are generally higher in altitude, sparsely populated, relatively backward in economic development, and have lower demand for ESs. The central and eastern regions are densely populated and had an overall higher level of economic development, which led to a higher level of land exploitation and, consequently, a higher demand for ESs. In 2000, the maximum value of ESs demand was only USD 10,190.8; in 2010, the maximum value reached USD 12,751.5, and by 2020, the value fell back to USD 12,248.9. The overall demand for ESs is fluctuating and rising. With the development of society, the level of land development and utilization and economic density have gradually increased, bringing about a rapid rise in demand for ESs. The lowest value of ESs demand was 0, which was mainly due to a lack of human activities in some NRs. 

### 3.2. Analysis of Ecosystem Service Supply and Demand Matching and Coupling Coordination Degree

Based on the S&D of ESs in NRs, this study matched the S&D for ESs using Equation (6) (Figure 4). Due to data limitations, there were 15, 17, and 18 NRs without data in 2000, 2010, and 2020, respectively. The H–H NRs were mainly concentrated in the eastern and central regions of China, which have a humid climate, more woodlands and wetlands in the region, and high ESs values provided per unit area. At the same time, due to the dense population and relatively high level of economic development, the demand for ESs is high. The H–L area is dominated by the northeast and southwest regions. The northeast NRs are influenced by temperate monsoons, with more wetland and woodland distribution, while the southwest region is under the influence of the subtropical monsoon climate and rugged terrain, with denser woodland distribution, which can provide higher ESs supply. However, human activities are low, the level of economic development is insufficient, and the demand for ESs is low. The L–L NRs were mainly distributed in the northeast and northwest. Some of the northeastern areas are affected by land desertification, and some NRs are sparsely vegetated, so the supply of ESs decreases; meanwhile, the northwest is mainly due to its inland location, lower rainfall, and more alpine meadows and deserts, and the value of the ESs provided is lower. At the same time, these areas are sparsely populated, with relatively low levels of economic development, and insufficient demand for Ess. The number of L–H NRs is low, mainly in the central and eastern parts of the country. It is mainly due to the high level of economic development dense population, and high land use in these areas, resulting in a limited supply of ESs in these NRs while the demand for ESs is high.

In terms of the number of NRs, the vast majority of NRs were dominated by H–H and L–L (Table 3). In 2000, H–H, H–L, L–L, and L–H NRs were 144, 99, 110, and 40 NRs, respectively, and by 2010, there were 155, 89, 100, and 47 NRs in H–H, H–L, L–L, and L–H NRs, respectively. The increase in H–H came mainly from H–L, with an increase of 16, while the increase in H–L came mainly from L-L, with an increase of 10. From 2010 to 2020, there was no change in the number of H–H NRs; L–H and L–L NRs decreased by 3 and 2, respectively, while 5 H–L NRs were added. The changed NRs were mainly focused on conversions between H–H and H–L.

Figure 5 reports the matching pattern of S&D for the nine types of NRs. The results show that the steppe meadow NRs from 2000 to 2010 were all in the third quadrant, and exhibited L–L. This may have been caused by the presence of low human activities in these NRs, where the LULC are mainly grassland and desert. By 2020, one of the NRs shifted from L–L to L–H, indicating that the demand for ESs in steppe meadow NRs was enhanced but the supply of ESs was not significantly enhanced. Geological relics and ancient organism relics were mainly distributed in the third and fourth quadrants and are expressed as L–L and L–H. Ocean coastal was mainly distributed in the second and third quadrants, showing H–L and L–L. Ocean coastal is mostly saline and wetland, with low land use intensity, resulting in low demand for ESs. Desert ecosystem was located in the third quadrant, i.e., L–L, for all years. The desert ecosystem NRs aim to protect deserts and reduce degradation and are mostly located in the northwest inland area. There are many deserts and grasslands, and the natural environment is harsh and unfavorable to human survival, resulting in a low level of S&D. Inland wetland and forest ecosystems were mostly distributed in the first and second quadrants, with higher levels of ESs supply in H–H and H–L. It is not difficult to understand that these two types of NRs protect mainly wetlands and woodlands, and both wetlands and woodlands enhance higher ESs supply. Wildlife was more evenly distributed, showing no obvious clustering of supply and demand, while wild plants were mostly located in the first quadrant, showing the H–H phenomenon.

Using Equations (7)–(10), we measured the CCD of ESs S&D in NRs. In 2000, the CCD ranged from 0.02 to 0.85, with a mean value of 0.53, showing largely basic coordination. In 2010 and 2020, the coordination degrees of S&D of ESs were 0.55 and 0.57, respectively. Which indicates that since 2000, the ESs S&D in NRs have become more coordinated. The number of NRs that reached coordination (>0.5) in 2000 was only 263, accounting for 66.75%, while the number rose to 278 in 2020, accounting for 70.74%. In terms of spatial distribution (Figure 6), the overall CCD of ESs S&D showed a decreasing trend from the southeast to the northwest. The severely dysfunctional NRs were mainly concentrated along the Qinghai–Tibet Plateau, which was caused by the H–L of these NRs.

### 3.3. Kernel Density Analysis

Using Equations (11) and (12), we measured the kernel density analysis of CCD. Figure 7 shows the dynamic evolution of the CCD of ESs S&D in NRs from 2000 to 2020. The overall NRs and ESs supply showed a “single peak”, with no major changes in 2000 and 2010, and the peak in 2020 shifted to the left. This indicates that the number of NRs with improved ESs supply increased substantially in 2010–2020. The overall ESs demand in NRs showed a “double-peak” state, with peaks decreasing in 2000–2010 and 2010–2020 and a more obvious trailing phenomenon on the right. This indicates that the difference in ESs demand between different NRs is gradually increasing, and the difference in service demand between different NRs is gradually increasing. In terms of the overall CCD of NRs, the kernel density function was single-peaked. Compared with 2000, the kernel density in 2010 was significantly lower than that in 2000 in the interval below 0.4, and the peak also shifted to the right. By 2020, the peak had further shifted to the right, indicating an overall increase in the CCD of NRs.

From 2000 to 2020, the peaks of geological relic NRs decreased, and the trailing left and right were more obvious, indicating that the gap of CCD within geological relic NRs was increasing. The peak in 2020 decreased sharply, and the curve tends to flatten and trail to the right. Among the desert ecosystem and inland wetland NRs, the kernel density curve and peak in 2010 shifted to the right compared with 2000, and the overall level of CCD improved. Meanwhile, the peak in 2020 shifted to the left, and the kernel density in areas with high CCD was higher than that in 2010, which means that the CCD in some desert ecosystems and inland wetland NRs decreased significantly in this decade. The kernel density trends of the CCD of ESs S&D in steppe meadow, ocean coastal, forest ecosystem, wildlife, and wild plant type NRs all show a shift of the kernel density curve to the right as the number of years increases, indicating that the CCD of ESs S&D in these NRs is steadily improving.

## 4. Discussion

### 4.1. Ecosystem Services Supply and Demand Patterns and Coupled Coordination Characteristics

This study measured the S&D of ESs in Chinese national NRs based on LULC. The study found that the ESs supply and demand per unit of NRs were higher in the central and eastern regions and lower in the western regions, especially in Xinjiang and the Qinghai–Tibet Plateau, which is consistent with the ESV trend in China [36,51]. Natural and LULC conditions are considered to be the main influencing factors in the formation of this spatial pattern of ply [52]. The western region is located in the interior of the continent, with an arid climate, many grasslands and deserts, and a low value of ecosystem services per unit area. The eastern region has a monsoonal climate with high vegetation cover, wetlands, and woodlands; hence, the value of ecosystem services per unit area is also high. China’s population is unevenly distributed, and its regional economic development shows a gradual decline from east to west. This pattern is closely related to the spatial distribution of developed land, which is proportional to population density and economic density. The differences in economic development and population status across regions are key factors that determine the variations in demand for ecosystem services. From the S&D matching pattern, the regions with more obvious S&D imbalance are concentrated in Northeast China and Southwest China, mostly in the H–L state [53]. This is due to the rugged terrain and cold climate in these areas, the low human activity, and the existence of extensive woodlands and wetlands, which lead to a high supply of ecosystem services. Furthermore, the low level of economic development and low land use reduce the demand for ecosystem services. [54]. The spatial coherence of ecosystem services is often overlooked, and this lack of spatial coherence has been demonstrated in several cities and for ecosystem services such as air purification, carbon storage, carbon sequestration, and air cooling [55,56], and this is no exception within nature reserves. Most of the NRs are in a state of H–H or L–L, with a decrease in areas of negative change in S&D patterns, such as from H–L to L–L. The region of positive change increased, such as from L–L to H–L. The main reason for this improvement is that the implementation of the NRs policy and the improvement in ecosystem governance have improved the supply of ESs [57]. In fact, contradictions in the S&D of ESs are a common phenomenon over time [58] influenced by a variety of factors such as LULC, socio-economic development, population layout, and macro policies in the region over a certain period of time [59].

A quantitative assessment of the CCD of ESs S&D in NRs is important for exploring and balancing the interactions between ecosystems and socio-economic systems [25]. On average, the CCD of ESs S&D in NRs has improved, and the number of NRs reaching coordination (>0.5) has been increasing. It indicates that the overall level of coordination between S&D of ESs in NRs can be improved under the NR policy system [51,52]. The results of kernel density estimation show that the CCD of ESs S&D in steppe meadow, ocean coastal, forest ecosystem, wildlife, and wild plant type NRs has effective protection. The forest ecosystems in NRs show better protection, due to the distance from human activities and the influences of reforestation [16]. Wildlife and wild plant NRs have also proven to be effectively protected from a biodiversity perspective after the reduction of human interference [60]. However, not all types of NRs are able to converge on coordination. The peaks of ESs supply, ESs demand, and the CCD of ESs S&D all improved significantly with increasing years. The overall trend of the CCD of desert ecosystems and inland wetland types first rose and then decreased, but the decrease was not enough to offset the increase caused by the rise, and overall improvement was still achieved. Desert ecosystem NRs have more fragile ecosystems and are more vulnerable in a climate change environment [20]. The ancient biological relics and geological relic types of NRs showed a greater overall fluctuation. This may be due to the smaller numbers and smaller unit areas of these two types of reserves. In contrast, the level of coupling coordination for all other types of NRs improved significantly. This indicates that the responses of different types of NRs to the CCD of ESs S&D for different conservation purposes also differed significantly. This is due to the fact that under the existing NRs system, priority conservation of biodiversity, forests, and forest carbon stocks is often emphasized, while insufficient attention is paid to ESs [61]. This requires decision makers to formulate conservation policies scientifically and rationally when establishing NRs and to take into account ESs while protecting biodiversity.

### 4.2. Policy Implications

It is essential to establish an S&D relationship between ESs and human benefits [62]. Incorporating the assessment of ESs S&D into the construction of NRs as a basis for policy implementation can ensure the effectiveness of policy implementation [30]. Since there is significant heterogeneity in the ESs S&D of Chinese national NRs, it is crucial to develop conservation policies and LULC patterns that match different regions and types of NRs. For example, LULC development can be appropriately enhanced in NRs with H–L, optimizing LULC patterns and improving land use efficiency to promote a balance between ESs S&D [63]. For NRs with different conservation purposes, different conservation measures are adopted. For example, ancient organism relics and geological relic types of NRs need to strengthen the protection of ESs and enhance the supply of ESs. To improve the effectiveness of ESs conservation in different types of protected areas, it is necessary to conduct regular assessments of ESs in NRs, and to adjust conservation policies according to changes in conservation effectiveness [43].

LULC patterns and their changes have a direct impact on supply and human demand [57]. The rational degree of people’s LULC determines the coupling state of land resources and ESs [59]. If not used properly, it will lead to the degradation of land resources and ecosystem functions [64]. Grassland, water bodies and wetland can provide a high supply of Ess, with a lower land use intensity index and lower demand of Ess, which can effectively promote the coordinated development of Ess. As the LULC with the highest level of land use intensity, construction land contrasts other LULC. The rapid expansion of urbanization and overdevelopment of real estate in recent years have had a negative impact on vegetation restoration and ecological environment improvement, and NRs have also been inevitably affected [65,66]. The occupation of other LULC by construction land reduces the ply in NRs while increasing the Ess demand, thus exacerbating the imbalance between Ess S&D. Therefore, it is necessary to strictly limit the scale of construction land under the existing NRs protection mechanism. Rational use of land resources for areas available for development is required to ensure that the demand for Ess for recreation of residents can be met while ensuring habitat quality and promoting sustainable development of human–land systems [67,68]. The protection of grassland, water bodies, and wetland should be strengthened in areas where the CCD is low to avoid the decline of ESs supply due to land degradation [69,70,71]. Therefore, under the existing nature conservation system, it is still necessary to further improve the NRs protection policy according to local conditions, coordinate human needs with ecological protection, and establish a long-term mechanism for NRs protection.

### 4.3. Limitations and Prospects

This study quantified the CCD of ESs S&D and assessed the effectiveness of ESs conservation in Chinese NRs. However, there are still some shortcomings.

In the assessment of ESs supply, the simplification of 25 LULC to 7 reduced the complexity of ESs estimation, but it was essential to conduct statistical analyses [72]. Xie et al. [33] provided a method to rapidly assess the value of ESs in China. However, as human activities in NRs are restricted, the ESs S&D generated will change. Further research is needed to propose a methodology that is tailored to assess the CCD of ESs S&D in NRs.

## 5. Conclusions

Based on remote sensing data of LULC and socio-economic data in 2000, 2010 and 2020, this study measured the spatial and temporal changes of ESs supply, ESs demand, and CCD of ESs S&D in 411 Chinese national NRs. On this basis, the impact of land use changes on the CCD of ESs S&D was analyzed. The conclusions of the study are as follows:Overall, the supply of ESs in NRs improved significantly from 2000 to 2020. Both ESs supply and ESs demand in Chinese national NRs showed a spatial pattern of increasing from west to east. In terms of different land use types, woodlands and grasslands provided the highest ESs supply, with the sum of the two accounting for 61.92% of the total value of ESs in 2000. From 2000 to 2020, both ESs supply and ESs demand showed an increasing trend. The matching of ESs S&D showed significant spatial heterogeneity. Central and eastern NRs were dominated by H–H and L–H, while the northeast, northwest, and southwest regions were mostly H–L or L–L.Since 2000, the CCD of ESs S&D has improved, and the number of NRs reaching basic coordination (>0.5) has increased by 15 in the past 20 years. The CCD of ESs S&D in steppe meadow, ocean coastal, forest ecosystem, wildlife and wild plant type NRs improved significantly; ancient organism relic, desert ecosystem and inland wetland type NRs fluctuated and increased, while geological relic type NRs as a whole showed a small decreasing trend.

## Figures and Tables

**Figure 1 ijerph-20-04845-f001:**
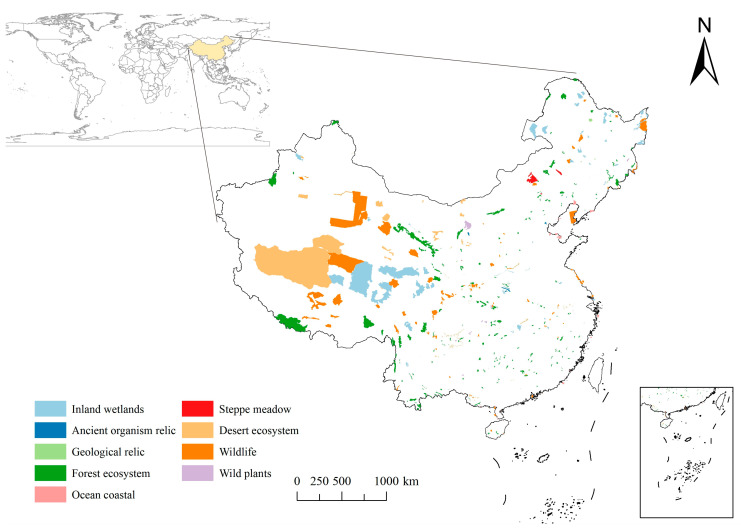
Distribution of National Nature Reserves in China.

**Figure 2 ijerph-20-04845-f002:**
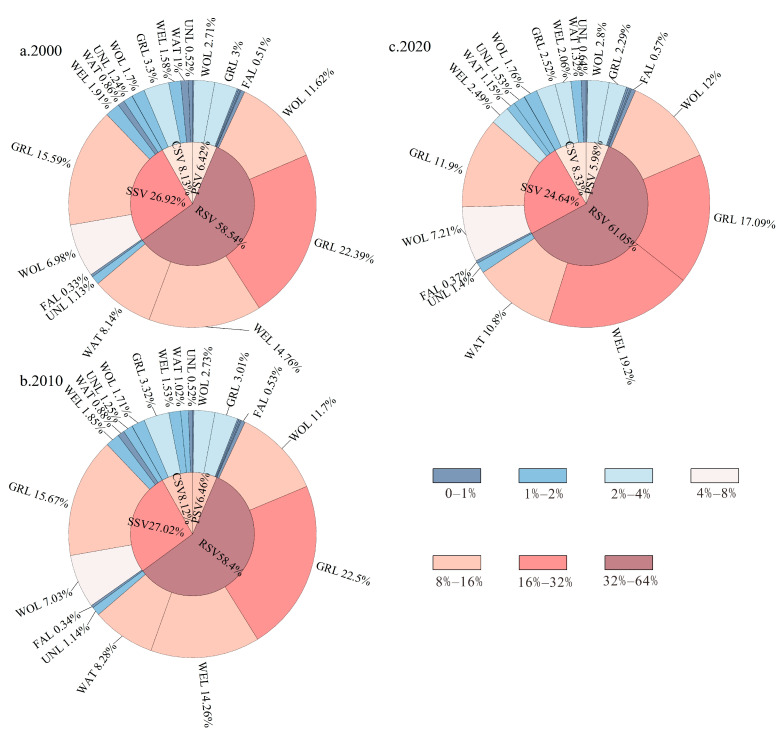
Share of land use type ecosystem services in total ecosystem services from 2000 to 2020. PSV: provision services values, RSV: regulate services values, SSV: support services values, CSV: cultural services values, FAL: farmland, WOL: woodland, GRL: grassland, WAT: water bodies, WEL: wetland, COL: construction land, UNL: unused land.

**Figure 3 ijerph-20-04845-f003:**
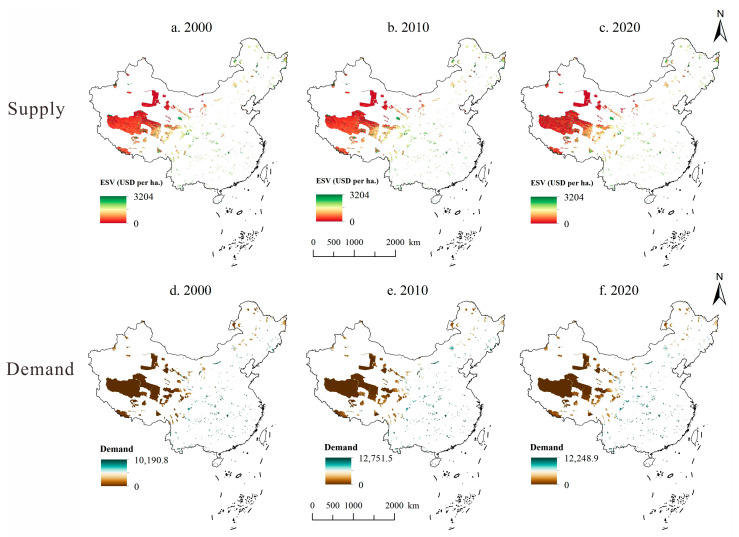
Ecosystem service supply and demand in National Nature Reserves from 2000 to 2020.

**Figure 4 ijerph-20-04845-f004:**
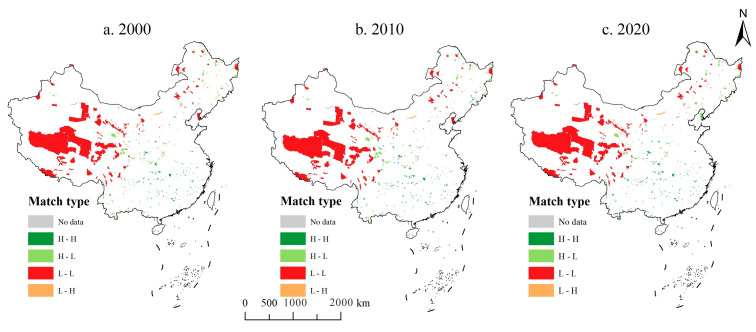
Ecosystem service supply and demand matching pattern in nature reserves from 2000 to 2020.

**Figure 5 ijerph-20-04845-f005:**
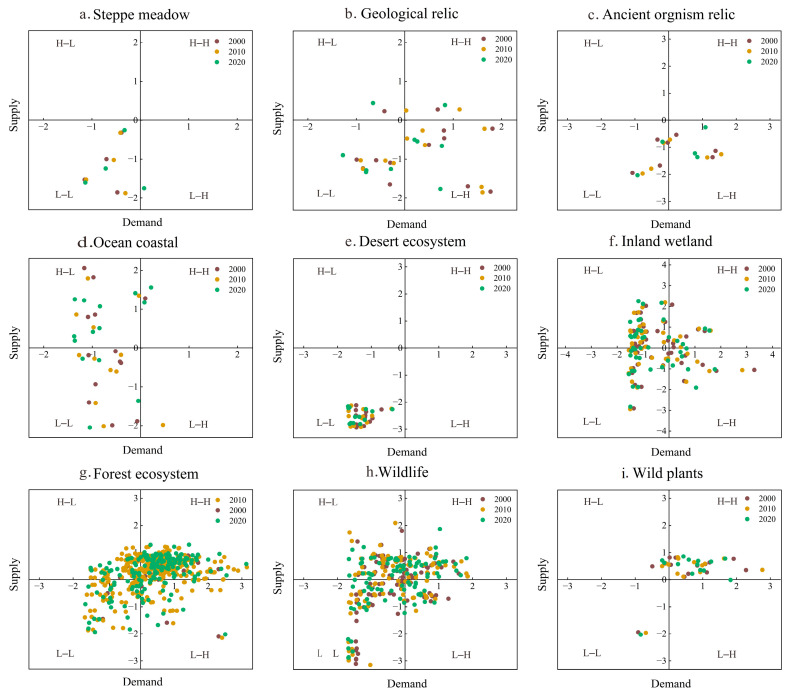
Ecosystem service supply and demand matching in different types of nature reserves from 2000 to 2020.

**Figure 6 ijerph-20-04845-f006:**
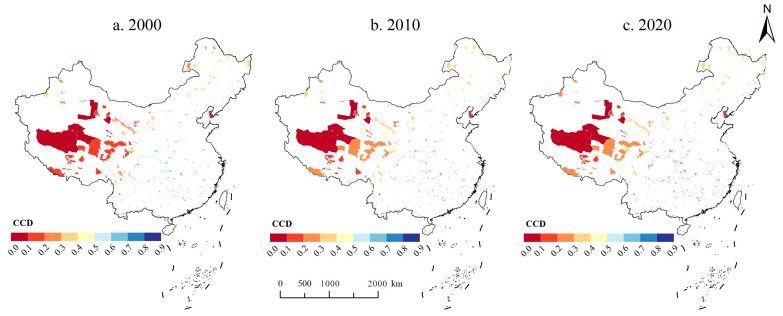
Ecosystem service supply and demand coupling coordination degree in National Nature Reserves from 2000 to 2020.

**Figure 7 ijerph-20-04845-f007:**
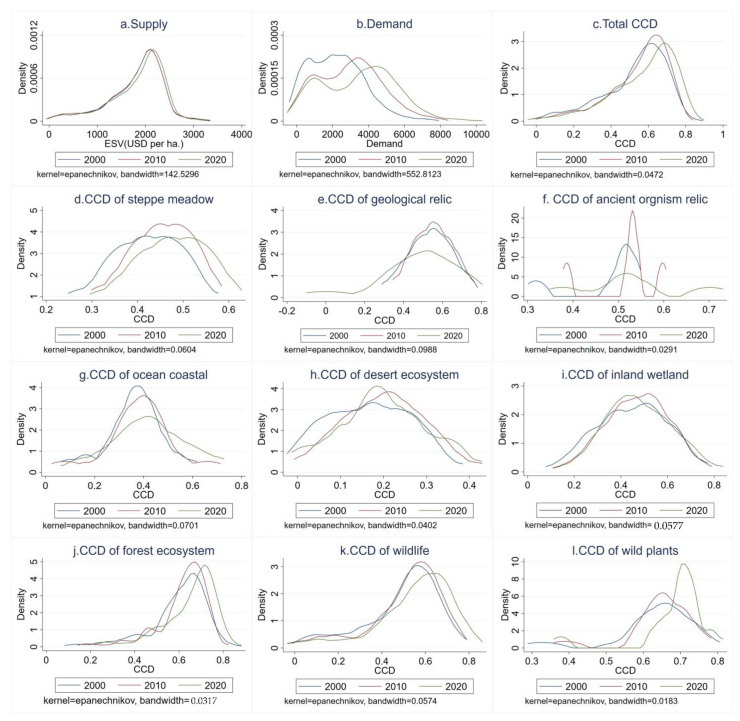
Kernel density estimation of ecosystem services supply–demand coupling coordination degree in different types of nature reserves from 2000 to 2020.

**Table 1 ijerph-20-04845-t001:** Land use intensity division in the study area.

Land Use Intensity Index (LDI)	Land Use Types	Land Use Intensity Index (LDI)	Land Use Types
LDI_1	Unused land	LDI_3	Farmland
LDI_2	Woodland	LDI_4	Construction land
	Grassland		
	Water bodies		
	Wetland		

**Table 2 ijerph-20-04845-t002:** Ecosystem services supply of land use types in nature reserves from 2000 to 2020 (USD millions).

	Year	FAL	WOL	GRL	WEL	WAT	UNL	COL	Total
PSV	2000	169.74	2501.40	2769.12	186.81	182.77	120.39	0.00	7930.23
	2010	174.91	2503.56	2765.21	179.36	184.74	120.46	0.00	7938.24
	2020	191.96	2610.22	2135.61	245.56	245.11	150.42	0.00	7598.88
RSV	2000	470.14	10,731.08	20,680.79	13,637.07	7516.61	1043.36	0.00	56,079.05
	2010	484.47	10,740.36	20,651.56	13,093.10	7597.48	1043.95	0.00	55,620.92
	2020	531.68	11,197.93	15,949.47	17,925.85	10,080.22	1303.64	0.00	59,008.79
SSV	2000	304.06	6446.20	14,406.45	1768.46	797.56	1143.68	0.00	26,866.41
	2010	313.33	6451.78	14,386.09	1697.92	806.14	1144.33	0.00	26,809.59
	2020	343.86	6726.64	11,110.56	2324.63	1069.58	1428.99	0.00	25,024.26
CSV	2000	20.76	1571.88	3049.54	1460.23	922.18	481.55	0.00	9506.14
	2010	21.39	1573.24	3045.23	1401.98	932.10	481.82	0.00	9465.76
	2020	23.48	1640.26	2351.87	1919.46	1236.70	601.68	0.00	9793.45

PSV: provision services values, RSV: regulate services values, SSV: support services values, CSV: cultural services values, FAL: farmland, WOL: woodland, GRL: grassland, WAT: water bodies, WEL: wetland, COL: construction land, UNL: unused land.

**Table 3 ijerph-20-04845-t003:** Transfer of matching supply and demand for ecosystem services from 2000 to 2020.

Time Interval	Match Type	Number of Nature Reserves			
		H–H	L–H	L–L	H–L
2000–2010	H–H	138	6	0	0
	L–H	16	82	1	0
	L–L	1	0	99	10
	H–L	1	0	2	37
2010–2020	H–H	140	9	0	6
	L–H	12	70	5	2
	L–L	0	7	88	5
	H–L	3	0	5	39

## Data Availability

The data presented in this study are available on request from the corresponding author.

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
