# Peer review of "Spatiotemporal Characteristics of the Coupled Coordination Degree of Ecosystem Services Supply and Demand in Chinese National Nature Reserves"

_ijerph, 2023, doi:10.3390/ijerph20064845_

Round 1
Reviewer 1 Report
This study analyzed the effectiveness of nature reserves and the supply and demand of ecosystem services in 412 Chinese Nature Reserves. The main result is that ecosystem services and nature reserves have improved.
I think that the idea of the article is interesting and worthy of publication in IJERPH.
In general, I think that the ms. is interesting, although it needs some improvement which will help make the message clearer. So, my recommendation is accepted after major revision.
Specific comments follow:
Title
Adequate and interesting.
Abstract
I found the abstract difficult to understand because of the overuse of acronyms (NRs, S&D, CCD). I think it would be clearer without these.
Introduction
Adequate and interesting. I found some typos in the text that need some English editing. Examples:
Line 34: I think the verb “realize” is not adequate here.
Line 38: It provides, instead of “It provide”
Line 51: “have been slowed down”… I think it is better to say: “has slowed down”
Line 67: Please explain what CCD means. Explain the acronym and then make sure what Coupling coordination degree refers to. This was not clear to me at all.
Methods
I understand the usefulness of calculating values in yuans. However, this is hard understand for people outside China. I would recommend using yuans AND another currency with which the public may be more familiar with. Example, line 145.
Also, can the authors improve the explanation of how to calculate the land use intensity index? (line 145)
Results
Table 1. The authors need to explain in the table heading what all the acronyms mean (FAL, WOL, GRL PSV, RSV etc… ) Otherwise, it is impossible to understand these results. These are explained in Figure 2, but still, it is necessary to add an explanation in the table.
Figure 2. Please, make the legends larger, and explain what the different colors mean.
Figure 5 provides little to no information. The authors should either improve this figure, or delete it. Do not keep it as is.
Figure 6. Although the authors mention H and L… these cannot be observed in the plots. This needs to be improved.
Figure 8. The axes need legends and units. Larger fonts would improve clarity. I had to enlarge the figure to 200% so that I would see it in detail.
Discussion
It needs to be more precise. For instance, what do the authors mean by “better natural ecology”? I do not understand this.
The authors seem to imply that an imbalance between S&D is bad. If supply is > than demand, it seems to me that it would be a good condition too (L 361, L 412).
L 367.- Please correct grammar here.
Sections 4.2 and 4.3 of the discussion seem a bit redundant. Maybe the authors can condense these two sections into one section?
Line 444. Please explain how are human activities different in NRs in comparison with the “outside world”.
Overall, the authors mention that ES S&D and NRs increase from west to east…I think this trend needs a better explanation. What happens in the country to have these patterns? This would be very interesting.
The results with H and L are difficult to see in the graphs and plots (they do not exist. This needs to be addressed.
The discussion would improve a great deal if the authors contrasted their results with other similar studies performed in China and elsewhere.
References
I think there are too many Chinese studies and only 2 from elsewhere. The authors could read and cite studies from different authors.
Author Response
Response to Reviewer 1 Comments
Dear Reviewer:
We are grateful to the reviewers for their valuable comments, which have been helpful in improving the quality of our manuscript. We have made all of the corrections suggested by you. We have responded to your suggestions and listed the page and line numbers for all related revisions in the manuscript. Please find our responses (in red font) to your comments and suggestions (in black font) below: Note that the revisions in the manuscript are highlighted in blue.
Thanks again for your review.
Best wishes,
Your Suggestions:
Point 1: I found the abstract difficult to understand because of the overuse of acronyms (NRs, S&D, CCD). I think it would be clearer without these.
Response 1: Thank you very much for your helpful suggestions. According to your comments, we have changed the abbreviations to full names in the abstract section to improve readability. (Line 13-31)
Point 2: Adequate and interesting. I found some typos in the text that need some English editing. Examples:
Line 34: I think the verb “realize” is not adequate here.
Line 38: It provides, instead of “It provide”
Line 51: “have been slowed down”… I think it is better to say: “has slowed down”
Response 2: Thank you very much for your professional opinion. We have made further polishing to the English grammar of the article. At the same time, we try to avoid non-standard academic expression.
Point 3: Line 67: Please explain what CCD means. Explain the acronym and then make sure what Coupling coordination degree refers to. This was not clear to me at all.
Response 3: Thank you very much for your professional opinion. Based on your comments, we have explained the CCD. (Line 90-92)
Point 4: I understand the usefulness of calculating values in yuans. However, this is hard understand for people outside China. I would recommend using yuans AND another currency with which the public may be more familiar with. Example, line 145.
Response 4: Thanks for the comments. We have changed the yuan to CNY. (Line 178)
Point 5: Also, can the authors improve the explanation of how to calculate the land use intensity index? (line 145).
Response 5: We sincerely appreciate the valuable comments. We have classified and tabulated the land use intensity of different land use types with reference to previous studies (Line 169, Table 1.)
Point 6: Table 1. The authors need to explain in the table heading what all the acronyms mean (FAL, WOL, GRL PSV, RSV etc… ) Otherwise, it is impossible to understand these results. These are explained in Figure 2, but still, it is necessary to add an explanation in the table.
Response 6: Thank you very much for your helpful suggestions. Based on your comments, we added an explanation of the abbreviations under Table 1. (Line 260-262).
Point 7: Figure 2. Please, make the legends larger, and explain what the different colors mean.
Response 7: Thank you very much for your suggestions. We made appropriate changes to Figure 2 and added a legend. (Line 263)
Point 8: Figure 5 provides little to no information. The authors should either improve this figure, or delete it. Do not keep it as is.
Response 8: We sincerely appreciate the valuable comments. We changed Figure 5 to Table 3 for a more detailed representation. (Line 330)
Point 9: Figure 6. Although the authors mention H and L… these cannot be observed in the plots. This needs to be improved.
Response 9: Thank you very much for your helpful suggestions. We numbered all the vignettes in Figure 6 and distinguished the supply and demand relationships corresponding to the four quadrants. (Line 352)
Point 10: Figure 8. The axes need legends and units. Larger fonts would improve clarity. I had to enlarge the figure to 200% so that I would see it in detail.
Response 10: Thank you very much for your helpful suggestions. Based on your comments, we added axes and units to Figure 8, and zoomed in on the details. (Line 395)
Point 11: It needs to be more precise. For instance, what do the authors mean by “better natural ecology”? I do not understand this.
Response 11: Thank you very much for your helpful suggestions. We have rewritten this sentence to get a more precise expression. (Line 416-418)
Point 12: The authors seem to imply that an imbalance between S&D is bad. If supply is > than demand, it seems to me that it would be a good condition too (L 361, L 412).
Response 12: Thanks for the comment. As you mentioned, it is a good condition if supply is > than demand. We have removed and rewritten the inappropriate statements.
Point 13: L 367.- Please correct grammar here.
Response 13: Thanks for the comment. We have made further polishing to the English grammar of the article.
Point 14: Sections 4.2 and 4.3 of the discussion seem a bit redundant. Maybe the authors can condense these two sections into one section?
Response 14: Thank you very much for your helpful suggestions. As you said, there is a duplication between 4.2 and 4.3, we have incorporated 4.2 into 4.3 and restated it. (Line 471-482)
Point 15: Line 444. Please explain how are human activities different in NRs in comparison with the “outside world”.
Response 15: Thank you very much for your helpful suggestions. We have rewritten the passage to make a distinction between human activities inside and outside the protected area. (Line 498-499)
Point 16: Overall, the authors mention that ES S&D and NRs increase from west to east…I think this trend needs a better explanation. What happens in the country to have these patterns? This would be very interesting.
Response 16: Thank you very much for your helpful suggestions. Based on your comments, we discuss and analyze the spatial patterns of supply and demand for ecosystem services. (Line 403-414)
Point 17: The results with H and L are difficult to see in the graphs and plots (they do not exist. This needs to be addressed.
Response 17: Thank you very much for your helpful suggestions. We have added Table 3 and optimized Figure 5, which will help to clarify the changes in ecosystem service supply and demand.
Point 18: The discussion would improve a great deal if the authors contrasted their results with other similar studies performed in China and elsewhere.
Response 18: Thank you very much for your helpful suggestions. We have added a comparison with other studies in the discussion section. (Line 400-403, Line 419-422, Line 439-441)
Point 19: I think there are too many Chinese studies and only 2 from elsewhere. The authors could read and cite studies from different authors.
Response 19: Thank you very much for your helpful suggestions. We reviewed literature from different backgrounds during the revision process and made references in the text.
Reviewer 2 Report
This is an interesting study is certainly worth evaluating. However, there are some key points that the authors should address prior to publications.
1. Most important of all, the reveiwer believes that the abstract does not really discusses the aims and background well. The authors are suggested to enhance the abstract.
2. Similar to the abstract, the introduction should also be enahnced. For instance, the past research on the same topic should be reported.
3. Conclusion section should be expanded with both qualitative and quantitative means.
4. Please enhance the quality of images (e.g., figure 8 should be subcategorized into a, b, c etc.)
Author Response
Dear Reviewer:
We are grateful to the reviewers for their valuable comments, which have been helpful in improving the quality of our manuscript. We have made all of the corrections suggested by you. We have responded to your suggestions and listed the page and line numbers for all related revisions in the manuscript. Please find our responses (in red font) to your comments and suggestions (in black font) below: Note that the revisions in the manuscript are highlighted in blue.
Thanks again for your review.
Best wishes,
Your Suggestions:
Point 1: Most important of all, the reveiwer believes that the abstract does not really discusses the aims and background well. The authors are suggested to enhance the abstract.
Response 1: Thank you very much for your helpful suggestions. We have restated the background of the abstract section and strengthened the study objectives. (Line 13-14,29-31)
Point 2: Similar to the abstract, the introduction should also be enahnced. For instance, the past research on the same topic should be reported.
Response 2: Thank you very much for your professional opinion. We have enhanced the literature review of previous studies and reported on the progress of relevant research in nature reserves. (Line 53-64)
Point 3: Conclusion section should be expanded with both qualitative and quantitative means.
Response 3: Thank you very much for your professional opinion. Based on your comments, we have optimized and added parts to the conclusion section. (Line 507-511)
Point 4: Please enhance the quality of images (e.g., figure 8 should be subcategorized into a, b, c etc.)
Response 4: Thanks for the comments. We have re-processed the images to improve their quality. (Line 263, 352, 395)
Reviewer 3 Report
It was a pleasure to review the paper "Spatiotemporal characteristics of the coupled coordination degree of ecosystem services supply and demand in Chinese national nature reserves", number " ijerph-2217462 " , submitted for publication in " International Journal of Environmental Research and Public Health (IJERPH)"
The paper is original, interesting and well written. Nonetheless, there are some improvements that should be done to improve the overall level of this manuscript, here are some comments:
1. Line 18, maybe it's better to use the past tense.
2. Please consider adding more results (numbers) in your abstract.
3. Line 23, "an" scientific basis ?
4. Line 30, please put a comma "," after 2018
5. Please write Km2 instead of Km2
6. You started to write about Land Use and Land Cover in lines 70-71, please try to add a more general background about it before this paragraph. Please also cite these 3 papers regarding LULC:
*https://doi.org/10.5194/essd-13-3907-2021
*https://doi.org/10.3390/land10040434
*https://doi.org/10.1038/s41597-022-01204-w
7. Figure 1, please consider including a map of the world, because it's hard to be located for non-chinese readers, the presence of the square is also not clear.
8. Please refer (in the text) to all equations used, according to equation number XX, or following equation number XX or (equation XX)...etc. for all of them.
9. Figure 8, please provide an X and Y axis in all your figures.
10. Honestly, your work in general and specifically your discussion was well handled. However, please try to focus more on your own results and compare them with other studies.
11. Please consider improving the overall language of your manuscript, there are some sentences that doesn't read well.
12. Please refer to the instructions for authors that can be found in the journal website, specifically regarding author contributions.
I wish the authors good luck.
Author Response
Dear Reviewer:
We are grateful to the reviewers for their valuable comments, which have been helpful in improving the quality of our manuscript. We have made all of the corrections suggested by you. We have responded to your suggestions and listed the page and line numbers for all related revisions in the manuscript. Please find our responses (in red font) to your comments and suggestions (in black font) below: Note that the revisions in the manuscript are highlighted in blue.
Thanks again for your review.
Best wishes,
Your Suggestions:
Point 1: Line 18, maybe it's better to use the past tense.
Response 1: Thank you very much for your helpful suggestions. We have made further polishing to the English grammar of the article. At the same time, we try to avoid non-standard academic expression.
Point 2: Please consider adding more results (numbers) in your abstract.
Response 2: Thank you very much for your professional opinion. We have refined the summary and added some numbers. (Line 22-28)
Point 3: Line 23, "an" scientific basis ?
Response 3: Thank you very much for your helpful suggestions. We refined the details and changed "an" to "a". (Line 29)
Point 4: Line 30, please put a comma "," after 2018
Response 4: Thanks for the comments. We have refined the details and added ",". (Line 38)
Point 5: Please write Km2 instead of Km2.
Response 5: We sincerely appreciate the valuable comments. We have changed the km2 in the full text to km2.
Point 6: You started to write about Land Use and Land Cover in lines 70-71, please try to add a more general background about it before this paragraph. Please also cite these 3 papers regarding LULC:
*https://doi.org/10.5194/essd-13-3907-2021
*https://doi.org/10.3390/land10040434
*https://doi.org/10.1038/s41597-022-01204-w
Response 6: Thank you very much for your helpful suggestions. We have carefully read and cited three excellent articles on land use that you have recommended to further strengthen the context of the introduction. (Line 46-49, 130-133, References 6, 7, 34)
Point 7: Figure 1, please consider including a map of the world, because it's hard to be located for non-chinese readers, the presence of the square is also not clear.
Response 7: Thank you very much for your suggestions. We have refined Figure 1 by adding the location of China in the world map. (Line 120)
Point 8: Please refer (in the text) to all equations used, according to equation number XX, or following equation number XX or (equation XX)...etc. for all of them.
Response 8: We sincerely appreciate the valuable comments. We show the references to equations in the results section of the text. (Line 238-239, 279, 298, 355,369)
Point 9: Figure 8, please provide an X and Y axis in all your figures.
Response 9: Thank you very much for your helpful suggestions. We added axes and units to Figure 8, and zoomed in on the details. (Line 395)
Point 10: Honestly, your work in general and specifically your discussion was well handled. However, please try to focus more on your own results and compare them with other studies.
Response 10: Thank you very much for your helpful suggestions. We have added a comparison with other studies in the discussion section. (Line 400-403, Line 419-422, Line 439-441)
Point 11: Please consider improving the overall language of your manuscript, there are some sentences that doesn't read well.
Response 11: Thank you very much for your professional opinion. We have made further polishing to the English grammar of the article. At the same time, we try to avoid non-standard academic expression.
Point 12: Please refer to the instructions for authors that can be found in the journal website, specifically regarding author contributions.
Response 12: Thanks for the comment. We have made appropriate revisions to the author contributions section, taking into account the official documents. (Line 517-521)
Reviewer 4 Report
The paper is well written and structured. The methodology is clear and concise. The literature review is also comprehensive. There are however, some minor deficiencies and grammar mistakes, which could be rectified promptly. Overall, this research adds a value to the academic community who are working in such domain.
1. The unit of measurement is not very standard. For example, the square (2) of square kilometers (km2) in the 32nd line should be superscript. The same problem occurs many times in the text.
2. The calculation of ecosystem service supply in this paper is based on the value coefficient method of Xie Gaodi, and has been revised using NDVI dataset. However, the revised coefficient is not shown in the text. Please show it in the later modified version.
3. In the 3.1 section of the results, the value of ecosystem services is sometimes introduced in billions USD sometimes in millions USD. It is suggested to unify so that readers can make an overall comparison.
4. The quality of Figure 2 is poor, and the content expressed is difficult to read. It is recommended to use professional drawing software such as origin for high-quality drawing.
5. In section 3.2 of the results, it is suggested to use numbers to classify the matching content of ecosystem service supply and demand, so as to obtain the matching features more easily.
6. In line 311, the expression of 2010 and 2020 lacks the preposition "in".
Author Response
Dear Reviewer:
We are grateful to the reviewers for their valuable comments, which have been helpful in improving the quality of our manuscript. We have made all of the corrections suggested by you. We have responded to your suggestions and listed the page and line numbers for all related revisions in the manuscript. Please find our responses (in red font) to your comments and suggestions (in black font) below: Note that the revisions in the manuscript are highlighted in blue.
Thanks again for your review.
Best wishes,
Your Suggestions:
Point 1: The unit of measurement is not very standard. For example, the square (2) of square kilometers (km2) in the 32nd line should be superscript. The same problem occurs many times in the text.
Response 1: Thank you very much for your suggestions. We have changed the km2 in the full text to km2
Point 2: The calculation of ecosystem service supply in this paper is based on the value coefficient method of Xie Gaodi, and has been revised using NDVI dataset. However, the revised coefficient is not shown in the text. Please show it in the later modified version.
Response 2: Thank you very much for your professional opinion. In this study, we performed the NDVI coefficient revisions based on the image element as a unit, which means that each image element possesses a different revision coefficient, so it is hard to show the NDVI coefficient results, but we provide a textual description of the range of NDVI revision coefficients. (Line 150-151)
Point 3: In the 3.1 section of the results, the value of ecosystem services is sometimes introduced in billions USD sometimes in millions USD. It is suggested to unify so that readers can make an overall comparison.
Response 3: Thank you very much for your suggestions. We have standardized the units of measurement in the text to millions USD. (Line 241-242)
Point 4: The quality of Figure 2 is poor, and the content expressed is difficult to read. It is recommended to use professional drawing software such as origin for high-quality drawing.
Response 4: Thank you very much for your suggestions. We modified Figure 2 based on origin software by enlarging the details of the image and adding a legend. (Line 263)
Point 5: In section 3.2 of the results, it is suggested to use numbers to classify the matching content of ecosystem service supply and demand, so as to obtain the matching features more easily.
Response 5: We sincerely appreciate the valuable comments. We changed Figure 5 to Table 3 for a more detailed representation. (Line 327)
Point 6: In line 311, the expression of 2010 and 2020 lacks the preposition "in".
Response 6: Thank you very much for your helpful suggestions. We have added the preposition "in" before 2010 here. (Line 357)